# Antibacterial Agents Adsorbed on Active Carbon: A New Approach for *S. aureus* and *E. coli* Pathogen Elimination

**DOI:** 10.3390/pathogens10081066

**Published:** 2021-08-22

**Authors:** Ewa Burchacka, Katarzyna Pstrowska, Elżbieta Beran, Hanna Fałtynowicz, Katarzyna Chojnacka, Marek Kułażyński

**Affiliations:** Department of Chemistry, Wroclaw University of Science and Technology, 50-370 Wroclaw, Poland; katarzyna.pstrowska@pwr.edu.pl (K.P.); elzbieta.beran@pwr.edu.pl (E.B.); hanna.faltynowicz@pwr.edu.pl (H.F.); katarzyna.chojnacka@pwr.edu.pl (K.C.); marek.kulazynski@pwr.edu.pl (M.K.)

**Keywords:** adsorption, pathogen elimination, active carbon, increased antipathogenic activity, antibiotics

## Abstract

Antibiotic overuse and mass production have led to a global problem with the treatment of antibacterial infections. Thus, any possibility to limit the number of antibacterial drugs used will contribute to a decrease in the development of pathogenic bacterial resistance. In this study, the enhanced bacterial growth reduction of pharmaceutical activated carbon (PAC) material with adsorbed antimicrobial agents compared to the activity of pure antibacterial drugs was investigated. Sulfamethoxazole (SMZ) at a concentration of 1.1 mg/mL retained the growth of *S. aureus* and *E. coli* at 20.5% and 26.5%, respectively, whereas SMZ adsorbed on PAC increased the reduction of the tested bacteria in the range of 47–72%. The use of PAC with adsorbed gentamycin (G) over 24 h improved the effectiveness of *E. coli* growth reduction by 50% compared to the application of pure antibiotic (3.6 µg/mL). The increased reduction of *S. aureus* growth by 6% using G with PAC for a 24-h incubation time compared to the use of pure antibiotics at a concentration of 3.6 µg/mL was observed. The results provide proof-of-principle that the new approach of activated carbon with adsorbed antimicrobial agents could yield an attractive background with potential as a new starting material for *S. aureus* and *E. coli* pathogen elimination, e.g., in wound-healing treatment in the future.

## 1. Introduction

Modern medicine is struggling with a very serious problem: the increasing resistance of bacteria to drugs that are used for antibiotic therapy. To prevent infection with pathogens that have developed resistance, doses higher than the recommended concentration of the selected drug are often needed [1,2]. This phenomenon is increasing significantly due to the widespread use of antibiotics on animal farms and the inadequate intake of antibiotics and nonoptimal antibiotic therapy [3]. These days, the evolution of microbial pathogens able to resist antibiotic treatment is seen as one of the most pressing public health crises (Table 1) [4,5]. Thus, any possibility of limiting the load of antibacterial drugs can contribute to reducing the development of resistance among bacteria.

Activated carbons (ACs) are on the World Health Organization’s list of essential medicines and contribute to treating diarrhea, indigestion, and flatulence [7]. Additionally, the removal of toxins from the skin as a result of the action of activated carbons contributes to their use in the cosmetic industry [8,9,10]. ACs are characterized by a high adsorption capacity, a well-developed and easy-to-modify surface, a high chemical stability, and mechanical strength [11]. ACs are modified to obtain substantially large surface areas (1 g of ACs can have a surface area over 3000 m^2^), which is achieved by their well-developed microporosity (pore size < 2 nm) [12,13,14]. Preparing ACs requires the use of a carbonization process and carbon activation. The activation process can be carried out by physical and chemical techniques. Activators such as H_3_PO_4_, ZnCl_2_, KOH, NaOH, and K_2_CO_3_ are commonly used for chemical activation [15,16,17]. Physical activation is done using gases such as carbon dioxide, air, or steam, usually at a temperature range of 600–900 °C [18,19,20].

ACs can adsorb pathogens, toxins, and spores, which extends their use in the treatment of subcutaneous tissue injuries and gastrointestinal infections. Adsorption tests of Gram-negative bacteria such as *Escherichia coli* on commercially available ACs have shown that the sorption capacity of bacteria increased with increasing the hydrophobicity and volume of macropores in the ACs [21]. Moreover, activated carbons that contain more oxygen species on their surface promote a higher level of adsorption of Gram-negative and Gram-positive bacteria and improve the weight and feed conversion ratios of broiler chickens when added to their feed [22]. The ability of ACs to adsorb many toxins, such as toxic metal ions, bilirubin, and protein-bound uremic toxins, has been demonstrated in scientific papers [23,24,25]. 

The use of bandages for wounds is necessary for proper healing. Commercially available dressings are characterized by a lack of porosity and adhesion to the wound surface [26] or a lack of proper therapeutic effect, which is a challenge for wound healing. To minimize the unpleasant symptoms of skin damage, many modern wound dressings provide several different functions, such as absorbing excess fluid or adsorbing unpleasant odors (e.g., Carboflex^®^, ConvaTec, Carbonet, Smith+Nephew, Clinisorb^®^, and Clinimed Ltd.). Commercially available active carbon bandages without an antibacterial agent (Bauer Bandage and KoCarbon) adsorb bacteria and odors, but their therapeutic applications are limited. Ag-activated charcoal dressings (ACTISORB™Silver 220, ACTISORB PLUS 25, Vliwaktiv Ag, and NOBACARBON-Ag), which absorb endotoxins and exhibit antibacterial properties, have a desirable effect on wound treatments. Enhancing wound healing by using antibacterial molecules (such as Ag) on the surface of active carbon bandages was clinically proven [27]. 

A similar strategy of using active carbon as a potential antibacterial component of bandages was adopted in this paper. This study investigates the effect of the antibacterial properties of active carbon with adsorbed antibacterial agents, such as gentamycin and sulfamethoxazole, which are used commercially in the form of ointments/creams to treat skin and soft tissue infections. Both antibacterial agents show the differences in their chemical structures and mechanisms of action. Gentamicin is a bactericidal aminoglycoside that binds to 16s rRNA on the 30-s ribosomal subunit. This leads to a disruption of the mRNA translation and, thus, leads to the formation of truncated or nonfunctional proteins [28]. Sulfamethoxazole is a sulfonamide bacteriostatic agent that is most commonly used in combination with trimethoprim as the drug. Sulfamethoxazole competitively inhibits dihydropteroate synthase, preventing the formation of precursor of folic acid, which is required for bacterial growth [29]. Studies on the gentamycin and sulfamethoxazole sorption ability of ACs indicated adsorption levels of around 2.5 and 0.25 g/g, respectively [30,31,32]. A comparison of the adsorption levels relative to the aminoglycoside and sulfonamide groups of antibiotics has never been carried out. Thus, gentamicin and sulfamethoxazole as representatives of the aminoglycoside and sulfonamide groups were selected for this study. The finding of an increased level of antibacterial activity of activated carbon with an adsorbed antibiotic can contribute to expanding the possibilities for antimicrobial therapies, including improving the conditions of infected wounds. Gram-positive pathogenic bacteria such as *Staphylococcus aureus* and *Streptococcus pyogenes* dominate in the early stage of the infectious process, and they are the most common causative agents of skin infection [33]. Several reports associated the Gram-negative enterobacterium *E. coli* with skin and soft tissue infections. *E. coli* was found to be the causative agent of neonatal omphalitis, cellulitis localized to the lower or upper limbs, necrotizing fasciitis, surgical site infections, and infections after burn injuries [34,35,36]. In this study, antibacterial activity was performed toward *S. aureus* and *E. coli* as representatives of Gram-positive and -negative pathogens presented in skin infections. 

Confirmation of the effectiveness of the constructed carbon material in both preliminary tests and later clinical trials could contribute to the production of new-generation bandages with antibacterial functions in the future.

## 2. Results

### 2.1. PAC Characterization

A commercial pharmaceutical active carbon (PAC) was used as the adsorbent. It contained a very low amount of ash, 0.80%, recalculated in a dry state (Table 2). This is typical for pharmaceutical ACs. According to the European Pharmacopoeia, the ash content should be lower than 5%. A low level of volatile matter (3.32% daf) indicates that the PAC has a stable structure that is not susceptible to thermal degradation. An analysis of the porous structure revealed that a majority of the PAC consisted of micropores (pores 0.4–2 nm wide), with a volume of 0.499 cm^3^/g.

The mesopore volume was 0.205 cm^3^/g, and the percentage share in the total pore volume was close to 30% (Table 2 and Figure 1). pH_PZC_ is the pH value at which the total charge of the adsorbent molecule is zero, and for the PAC, the pH_PZC_ was 7.01 (Table 2). Interactions with electron acceptors and cationic electron donors are preferred when the pH of the solution is greater than the pH_PZC_. However, when interacting with electron acceptors and anionic electron donors, they are preferred at a pH lower than the pH_PZC_ [37,38].

### 2.2. Elemental Content of PAC

The PAC revealed an oxygen content of around 5% on its surface and a chloride content above 0.2%. The contents of other elements, such as Na, Al., Si, P, S, K, and Ca, were imperceptible (<0.2%), and their summed content equaled 0.67% weight (Figure 2).

### 2.3. Adsorption of Antibacterial Agents on PAC

The adsorption of gentamycin on PAC was 36% during 5 min of contact, but after 6 h of contact, the sorption of G was increased to 56% (Figure 3). After 24 h of contact of the adsorber with the antibiotic, the level of gentamicin adsorption remained unchanged and was still 56%. In the case of sulfamethoxazole adsorption, regardless of the time of contact between PAC and the antibiotic, the level of adsorption was unchanged, amounting to 100% of the adsorbed sulfamethoxazole under the conditions of the test.

### 2.4. Adsorption Isotherms of Antibiotics on PAC

The equilibrium adsorption isotherms for gentamycin and sulfamethoxazole presented in Figure 4 display q, the amount of antibacterial agents adsorbed (g/g adsorbent), against Ceq, the residual concentration of gentamycin sulfate or sulfamethoxazole in the solution. The results indicated that PAC had an adsorption capacity of over 4 g/g for gentamycin sulfate, and that of sulfamethoxazole was around eight times lower (0.5 g/g). The sulfamethoxazole isotherm for PAC was characterized by a very rapid initial growth, followed by a (pseudo-)plateau, indicating a strong affinity of the antibiotic with the adsorbent [39]. The adsorption isotherm of sulfamethoxazole (SMZ; Figure 4a) on PAC at 24 °C during 6 h of incubation was hyperbolic and represented the best fit by the type I isotherm model [40,41]. The gentamycin isotherm is sigmoidal (type II isotherm), with the most steeply rising part being vertical, which usually suggests an ideally homogeneous surface (Figure 4b) [39]. In this case, the surface of PAC is known to be highly variable. The sigmoidal isotherm is expected to describe the adsorption of strongly polar adsorbates in the pores of a hydrophobic solid where adsorbate clustering or capillary condensation can occur.

### 2.5. Bacterial Growth Reduction

Sulfamethoxazole (SMZ) at a concentration of 1.1 mg/mL, which corresponded with a concentration of SMZ adsorbed on PAC, was used as the control in bacterial growth reduction studies. SMZ (1.1 mg/mL) retained the growth of *E. coli* and *S. aureus* at 26.5% and 20.5%, respectively (Figure 5, red, thick dashed line). The tested material—PAC with adsorbed SMZ—increased the reduction of *E. coli* and *S. aureus* by 35% (Figure 5B, parameter x) and 51% (Figure 5E, parameter x′), respectively, compared to the control sample. Additionally, the adsorption level of *E. coli* and *S. aureus* on PAC (without antibacterial agents) was examined, and the adsorption level was 40% (Figure 5A) and 27% (Figure 5D), respectively. The antimicrobial activity of gentamycin at concentrations of 3.6 µg/mL (which corresponded with a concentration of G adsorbed on PAC at 24 h) reduced the growth of *E. coli* by 40% (Figure 5, left, blue, thick dashed line). Gentamycin showed a much higher antibacterial activity toward Gram-positive bacteria such as *S. aureus*; at a concentration of 3.6 mg/mL, it reduced the growth of bacteria by 72% (Figure 5, right, blue, thick dashed line). The use of PAC with adsorbed gentamycin after 24 h of contact time improved the effectiveness of *E. coli* growth reduction by 50% compared to the use of pure antibiotic (3.6 µg/mL; Figure 5C, parameter y). A reduction of bacterial growth using PAC with adsorbed G was observed for *S. aureus* to a lesser extent. The reduction of *S. aureus* increased by only 6%, using the sorbent after 24 h of contact time with gentamycin compared to the pure antibiotic at a concentration of 3.6 µg/mL (Figure 5F, parameter y′).

The adsorption of SMZ to PAC increased the reduction of tested bacteria depending on the contact time (Figure 6, left). The application of the material after 5 min of adsorption produced the highest level of bacterial reduction: 72% in *S. aureus* (Figure 6A) and 62% in *E. coli* (Figure 6B). Surprisingly, extending the adsorption time of SMZ with PAC to 24 h decreased the reduction of the bacterial culture to 47% toward *S. aureus* (Figure 6A′). Extending the incubation time of sulfamethoxazole with PAC from 5 min to 24 h slightly decreased the *E. coli* growth from 62% to 54% (Figure 6B,B′). However, irrespective of the incubation time, the use of PAC with sulfamethoxazole significantly improved the bacterial reduction in the culture. Contrary to sulfamethoxazole, gentamycin was not 100% adsorbed on PAC; the concentration of adsorbed G was 3.6 and 2.3 µg/mL for 24 h and 5 min of incubation time, respectively. The G incubated for 5 min with PAC slightly decreased the growth of both *S. aureus* and *E. coli* to 67% (Figure 6C) and 72% (Figure 6E), respectively, compared to the material where the time of contact of G with PAC was 24 h (Figure 6D,F). However, the decreased concentration of adsorbed antibiotic (2.6 µg/mL) during the 5 min of incubation time with PAC revealed a decreased antimicrobial activity relative to the higher concentration (3.6 µg/mL) obtained with the longer (24 h) contact time of the absorber with G (Figure 6D,F).

### 2.6. Release of Antibacterial Agents from PAC

The mean pH value of chronic wounds was found to be 7.4 [42]. Our data indicate that sulfamethoxazole is not released from pharmaceutical-activated carbon for up to 24 h under aqueous conditions in pH 7.4 (Figure 7). The release of gentamycin sulfate from PAC under aqueous conditions was observed to a small extent (3%) after a few minutes; however, it was constant at that percentage up to 24 h (Figure 7).

## 3. Discussion

The molecular weight (253 g/mol) and size (0.929 nm diameter) of sulfamethoxazole are lower than those of gentamycin (477 g/mol, 1.148 nm diameter). The adsorption of organic compounds from dilute aqueous solutions can be affected by factors such as the molecular weight of the compound, the size and geometric shapes of the particles, the functional groups, polarity, solubility, and dissociation constant. Molecules are most readily adsorbed in pores that are two times wider than the longest spatial dimension of the compound [43]; thus, the sorption of sulfamethoxazole seems to take place mainly in micropores (which constitute about 50% of the PAC surface), whereas the sorption of gentamycin occurs in mesopores with a 2 to 3-nm width. Additionally, a lower solubility of SMZ in water (1 mg/mL) [44] as compared to gentamycin (100 mg/mL) [44] causes the easiest dissociation of G in water and, thus, a lower level of PAC sorption, which reveals that the increase in adsorption on activated carbon depends mainly on the solubility and size of the adsorbed molecule, not on electrostatic interactions. Thus, it was not surprising to observe a higher sorption level of SMZ on PAC compared to G (Figure 3).

Furthermore, 100% SMZ adsorption was determined immediately after contact of the sorbent with the substance, and this level was maintained for 24 h. Gentamycin was adsorbed via PAC at a level of 36% during 5 min of contact, but after 6 h of contact, the sorption increased by 20%. The 56% level of G adsorption on PAC remained stable for 24 h (Figure 3). The differences in the adsorption process, in this case, resulted from two factors: the electrostatic charge and the size of the adsorbed molecules. Effective SMZ adsorption was observed with the micropore sorbents (such as PAC), while effective G sorption can be achieved with mesopore materials (pores > 2 nm). Regardless of the geometric dimensions of the molecules, the significant adsorption observed in both cases was the result of the large PAC BET surface area (1267 m^2^/g; Table 2). The research planned for the future will take into account optimizing the amount of active carbon for the given solutions. The SMZ adsorption curve indicates that a smaller amount of carbon (or a higher concentration of SMZ) could be used for the adsorption process. Nevertheless, the optimization of this parameter was not the main goal of this paper and was omitted.

PAC shows a sorption of *E. coli* and *S. aureus* of 40% and 27%, respectively (Figure 5A,D). It seems that electrostatic interactions do not facilitate the adherence of bacteria to a carbon surface. The PAC surface remains negatively charged; thus, an electrostatic interaction with a cationic electron donor should be preferred. Generally, the bacterial cell wall has a negative charge. In Gram-positive bacteria such as *S. aureus*, the negative charge is due to the presence of teichoic acids linked to either the peptidoglycan or the underlying plasma membrane. These teichoic acids are negatively charged due to the presence of phosphate functional groups in their structures. Gram-negative bacteria such as *E. coli* have an outer covering layer of phospholipids and lipopolysaccharides. The latter imparts a strong negative charge to the surface of Gram-negative bacteria cells. It was previously reported that the elemental content and electron microscopy (EM) analysis of active carbons indicated that the oxygen content is important in the adsorption of Gram-positive and -negative bacteria over electrostatic interactions [22]. In the tested carbon material, PAC, the oxygen content was significant, as it amounted to 5%.

For both gentamicin and sulfamethoxazole, regardless of the time of incubation with the adsorber, an increased bacterial growth reduction against both Gram-positive and -negative bacteria was observed. The obtained results demonstrated that activated carbon supports the action of antibacterial/bacteriostatic agents toward bacterial growth reduction. The adsorption of the antibiotic to the PAC did not limit the interaction of the antibacterial substance with the target bacterial cells in the culture. Rather, the increase of bacterial growth reduction was observed. We hypothesized that the higher level of bacterial growth reduction in the culture was caused by the antibacterial/bacteriostatic properties of the organic substance adsorbed on PAC (no limit of the antibacterial agent interactions with bacteria) with a simultaneous adsorption process of bacteria on active carbon.

Extending the contact time of PAC with SMZ decreased the percentage reduction of bacteria such as *S. aureus* and *E. coli* (Figure 6A,A′,B,B′). Although SMZ was 100% adsorbed by PAC, the prolonged contact time most likely allowed it to adsorb to the micropores. Assuming that the SMZ was adsorbed in the micropores after 24 h of contact with PAC (not available for bacteria), it can be concluded that its contact with the bacterial cells was limited, decreasing the percentage reduction in bacterial growth in the culture. Our future research will focus on the explanation of the interaction of carbon with the antibacterial substance such as SMZ to exact an explanation of the mechanism of action of activated carbon with antibacterial substances toward the reduction of bacteria.

The 56% adsorbed G on PAC with 24 h of contact should showed a higher bacterial growth reduction than the 36% adsorbed with 5 min of contact. It was not a surprise that the prolonged contact time of PAC with G (24 h) increased the ability to reduce *S. aureus* and *E. coli* cells by ~10% and ~20%, respectively (Figure 6, right). However, the 24 h of contact time may additionally allow for an increase in the surface area of G deposition on the activated carbon, thus providing a greater surface area for contact between the antimicrobial substance and bacterial cells. It appears that the use of activated carbon together with an antimicrobial substance improves the bacterial growth reduction in the culture as a result of the physical trapping of the bacteria on the surface.

No release or a very limited release of active substances from activated carbon as a carrier is important from the point of view of limiting the amount of antibiotic adsorption during wound-healing treatments. At a pH similar to that of chronic wounds (7.4), no release of sulfamethoxazole and 3% release of gentamicin from PAC was observed. However, in vitro wound healing or animal model assays regarding the release of antibiotic adsorbed on PAC should be performed to confirm the obtained results.

The presented results are the beginning of research on the antibacterial activity of activated carbon with an adsorbed antibiotic with the possibility of using it in antibacterial bandages in the future. The subsequent research will expand to include quantitative research using SEM. Additionally, it is planned to perform bacterial viability tests to refine the obtained results of antibacterial activity. It will be necessary to test the antimicrobial properties under a variety of conditions (including those suitable for human skin). It is planned to conduct tests on activated carbons with different physicochemical properties to study the influence of the chemical composition on the ability to reduce bacteria and the level of adsorption of various antibacterial substances. Extending the scope of the research will thus allow the determination of the detailed mechanism of the process.

## 4. Materials and Methods

### 4.1. Characteristics of PAC

The sorbent used in this study was a commercially available pharmaceutical active carbon (PAC) of biomass origin (Aflofarm, Pabianice, Poland). The active carbon was characterized by a proximate analysis (moisture, ash, and volatile matter) in compliance with the Polish standards (PN-80/G-04511, 04512, and 04516). To determine the pH_PZC_ value of PAC, a protocol based on the ISO 787-9:1981 standard was used. First, 20 mL of distilled water was poured into a flask containing 0.5 g of AC, and the flask was tightly closed and shaken for 72 h at room temperature. The pH was measured in triplicate. The porous structure of PAC was determined on a gravimetric apparatus of the McBain–Bakr type at 25 °C, using carbon dioxide adsorption and C_6_H_6_ adsorption and desorption isotherms in the pressure range of 0–700 mmHg and the relative pressure range of p/po 0-1 (for benzene only). The sorption apparatus scheme was presented in our previous study [45]. The Dubinin–Radushkevich (DR) theory was used for interpreting the CO_2_ adsorption isotherms (the adopted affinity coefficient β was 0.37) [46]. The DR equation was used to calculate the volume of micropores available for carbon dioxide. The surface area of these pores was calculated with the assumption that 1 mole of CO_2_ at 25 °C lies flat on the surface of the micropores and covers an area of 0.185 nm^2^ [47]. The benzene adsorption isotherm was interpreted using the Brunauer, Emmet, and Teller (BET) theory. An isotherm range between the relative pressure p/po of 0.01 and 0.30 was used, and the assumption that C_6_H_6_ at 25 °C occupied an area of 0.41 nm^2^ was made [48,49]. The volume (VMES) and size distribution (SMES; 2–3, 3–5, 5–10, and 10–50 nm as a function of the width) of the mesopores were calculated based on the C6H6 desorption isotherm (p/po = 0.96–0.175) according to the Pierce method [50,51]. The micropore volume was calculated as the difference according to Equation (1):VMICB = VG − MES (cm^3^/g) (1)
where VG is the Gurvich volume, which is the volume of benzene adsorbed at a relative pressure of 0.96, and VMES is the calculated volume of the mesopores. The volume of the submicropores was calculated as the difference between the micropore volume calculated from the CO_2_ adsorption curve (VMICC) and from the benzene adsorption curve (VMICB):VSUB = VMICC − MICB (2)

The average mesopore width (dMES) was calculated from the equation: dMES = 2VMES/SMES(3)

### 4.2. Elemental Content of PAC

The elemental composition of pharmaceutical carbon was examined using a Tracor-Northern dispersion energy spectrum X-ray spectrometer (EDX) mounted on a Quanta 250 FEI scanning electron microscope operating at 15 kV. The elemental content results were presented as a percentage of the average weight value from 6 independent places on the surface of active carbon (% wt ± SD) (Appendix A).

### 4.3. Adsorption of Antibacterial Agents on PAC

#### 4.3.1. Adsorption of Gentamycin Sulfate (G) on PAC

The activated carbon (PAC; 20 mg) was measured into three Eppendorf tubes, and 1 mL of 6.4-µg/mL gentamycin solution was added. The samples were incubated at room temperature for 5 min, 6 h, and 24 h. The sample solutions were filtered by Whatmann^®^ filter paper to separate any hard suspended particles (such as carbon). Then, 50 mL of a 5-mg/mL ninhydrin solution was added to 150 mL of the obtained filtrate of gentamycin sulfate. The samples were vortexed, heated, cooled, and spectrophotometrically examined according to the procedure described in the Analysis section of the Appendix A. To determine the adsorption of gentamycin on activated carbon, a linear equation of the standard curve was used (Appendix A). Using the standard curve, the concentration of gentamycin sulfate left in the solution after adsorption was determined. The amount of gentamycin sulfate adsorbed on PAC was calculated as the difference between the initial (6.4 µg/mL) and final concentrations of gentamycin in the solution. All measurements were performed in duplicate. The results are presented as the percentage of adsorbed gentamycin sulfate ± SEM.

#### 4.3.2. Adsorption of Sulfamethoxazole (SMZ) on PAC

The activated carbon (PAC; 20 mg) was weighed into three Eppendorf tubes; 1 mL of 1.1-mg/mL sulfamethoxazole solution was added; and the mixture was incubated for 5 min, 6 h, and 24 h. After adsorption, the PAC was removed from the solution using Whatmann^®^ filter paper. The concentration of S in the solution after the adsorption was analyzed by HPLC (according to the procedure described in the Analysis section of the Appendix A). The calculation of the amount of non-absorbed sulfamethoxazole on the PAC was performed using the standard curve (Appendix A). The amount of sulfamethoxazole adsorbed on the PAC was calculated as the difference between the initial (1.1 mg/mL) and final concentrations in the solution. All the measurements were performed in duplicate. The results are presented as the percentage of adsorbed gentamycin sulfate ± SEM.

### 4.4. Adsorption Isotherms of Antibacterial Agents on PAC

#### 4.4.1. Adsorption Isotherms of Gentamycin Sulfate (G) on PAC

Different weights of PAC in the range of 0.1–50 mg were placed in tubes, and 1 mL of 6.4-mg/mL of G was added. The mixture was incubated for 6 h when the adsorption equilibrium was reached, with gentle shaking at room temperature. Centrifugation of the mixture at 5000 rpm for 15 min was performed to separate the liquid from the solid. The concentration of G in the supernatant samples was measured according to the procedure described in the Analysis section of the Appendix A.

#### 4.4.2. Adsorption Isotherms for Sulfamethoxazole (SMZ) on PAC

Different weights of PAC in the range of 0.6–22 mg were placed in tubes, and 1 mL of 1.1-mg/mL SMZ was added. Then, the mixture was incubated for 6 h with gentle shaking at room temperature. Centrifugation of the mixture at 5000 rpm for 15 min was used to separate the liquid from the solid. The supernatant samples were measured according to the procedure described in Analysis Section of the Appendix A.

### 4.5. Bacterial Growth Reduction by Adsorbed Gentamycin Sulfate (G) and Sulfamethoxazole (SMZ) on PAC

The ability of active carbon (PAC) to adsorb bacteria was assessed with the biosorption of the *E. coli* (ATCC 8739) and *S. aureus* (ATCC 6538) species, which were grown in LB broth (Miller) and Mueller Hinton (MH) broth, respectively, at a temperature of 37 °C. Overnight cultures of the tested bacteria were diluted 200-fold in fresh LB and MH medium, respectively, and incubated at 37 °C until the OD_600_ reached 0.3. Then, 1 mL of appropriate bacteria was added to 4 sterile tubes containing 20 mg of crude PAC (1 tube) and PAC with adsorbed gentamycin or sulfamethoxazole (3 tubes) by the method described in Section 4.3. The concentrations of gentamycin in the control sample were 2.3 and 3.6 mg/mL, and the concentration of the sulfamethoxazole control sample was 1.1 mg/mL. The investigated sample and control sample were stirred by vortexing for 2 min and then shaking at 37 °C (optimal growth temperature of the bacterial culture) with slight agitation (45 rpm). PAC with adsorbed G or SMZ and control samples (crude PAC and bacteria cultures) was filtrated using Whatmann^®^ filter paper. The absorbance results of the control samples before and after filtration were described in Appendix A. The absorbance was measured at 600 nm to determine the reduction of the bacterial cell growth. To determine the adsorption or antibacterial level of PAC with adsorbed gentamycin, the following equation was used:A% = 100% − ((OD600 of PAC or PAC+G/SMZ)/(OD600ofCON) × 100%) (4)
where PAC is the bacterial solution with active carbon, PAC+G/S is the bacterial solution with adsorbed gentamycin or sulfamethoxazole on active carbon, and CON is the control sample of bacteria (without the sorbent). All measurements were performed in duplicate. The results are presented as a percentage of the adsorbed bacteria ± SEM.

### 4.6. Release of Antibacterial Agents from PAC

#### 4.6.1. Release of Gentamycin Sulfate from PAC under Aqueous Conditions

A volume of 10 mL of 6.4-mg/mL G solution was added to 200 mg of activated carbon (PAC). The mixture was incubated for 6 h at room temperature and centrifuged for the removal of non-adsorbed antibiotics. Then, 10 mL of PBS (pH 7.4) was added to the remaining adsorbent, and 1 mL of the supernatant was withdrawn over 0, 2, 6, 10, 12, and 24 h. Then, 150 mL of the sample was added to 50 mL of a 5-mg/mL ninhydrin solution. The prepared samples were vortexed and heated at 95 °C for 5 min. After the elapsed time, the solutions were transferred to an ice bath for 1 min and immediately examined spectrophotometrically at 418 nm. All the measurements were performed in duplicate (Appendix A).

#### 4.6.2. Release of Sulfamethoxazole from PAC under Aqueous Conditions

Sulfamethoxazole (10 mL of 1.1 mg/mL) was added to 200 mg of activated carbon (PAC). The mixture was incubated for 6 h at room temperature and centrifuged for the removal of non-adsorbed antibiotics. Then, 10 mL of PBS (pH 7.4) was added to the remaining adsorbent, and 1 mL of supernatant was withdrawn over 0, 6, and 24 h. The samples were analyzed by high-performance liquid chromatography (Waters 2707 Instrument) according to the procedure described in the Analysis section of the Appendix A.

## 5. Conclusions

This work aimed to demonstrate the effect of activated carbon with adsorbed antibacterial agents in achieving an improved reduction of *S. aureus* and *E. coli* pathogen growth. The highest levels of *E. coli* (62%) and *S. aureus* (72%) reduction were achieved for the PAC that adsorbed SMZ after 5 min of contact. On the contrary, the contact of PAC with G over 24 h increased the *S. aureus* and *E. coli* growth reductions by 11% and 18%, respectively, compared to the PAC that adsorbed G after 5 min of contact. The commercialization of activated carbon with adsorbed antibacterial substances requires a pharmacological and clinical research application, but this work shows a new approach to a PAC-adsorbed antibacterial agent in terms of an enhanced pathogen grow reduction compared to the activity of pure antibacterial agents.

## Figures and Tables

**Figure 1 pathogens-10-01066-f001:**
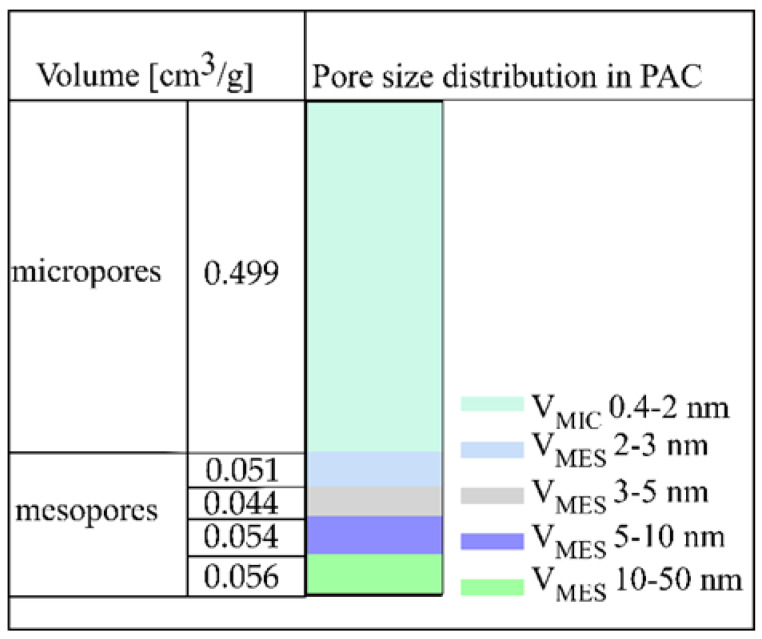
Pore size distribution in PAC.

**Figure 2 pathogens-10-01066-f002:**
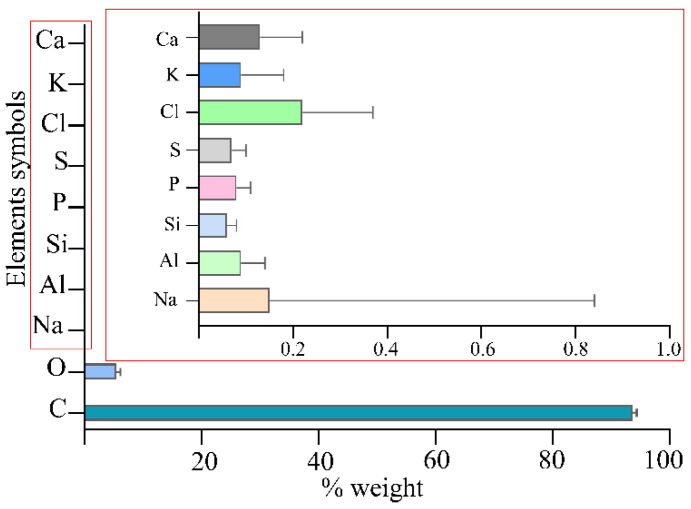
Elemental composition of PAC.

**Figure 3 pathogens-10-01066-f003:**
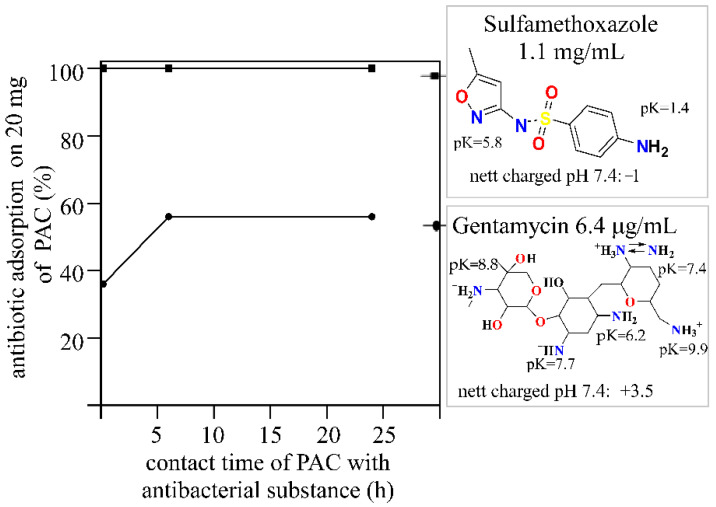
Sorption curves of gentamycin and sulfamethoxazole on PAC: influence of the contact time.

**Figure 4 pathogens-10-01066-f004:**
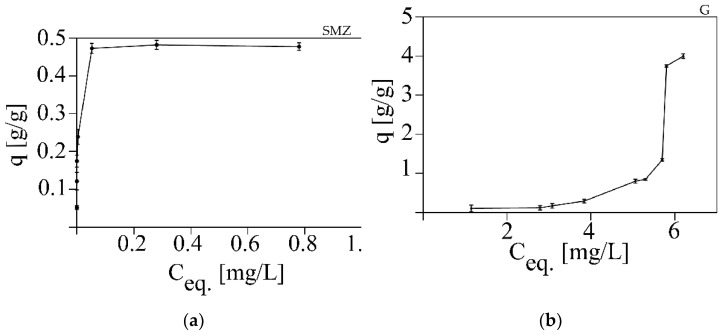
Adsorption isotherms of (**a**) sulfamethoxazole (SMZ) and (**b**) gentamycin (G) on PAC at 24 °C during 6 h of incubation.

**Figure 5 pathogens-10-01066-f005:**
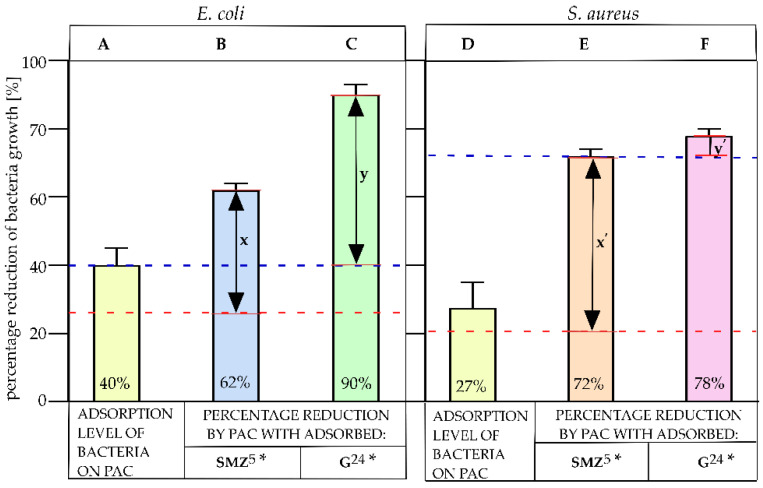
Percentage reduction of bacteria growth. Adsorption level of *E. coli* on PAC (**A**), *E. coli* growth reduction by PAC with the adsorbed SMZ (**B**), and *E. coli* growth reduction by PAC with the adsorbed G (**C**). Adsorption level of *S. aureus* on PAC (**D**), *S. aureus* growth reduction by PAC with the adsorbed SMZ (**E**), and *S. aureus* growth reduction by PAC with the adsorbed G (**F**). The bacteriostatic activity of SMZ in 1.1 mg/mL is marked with a red, thick dashed line. The antibacterial activity of G in 3.6 µg/mL is marked with a blue, thick dashed line. The concentrations of pure SMZ and G correlated with the concentration of those substances on PAC. The double arrow with x and x′ indicates the percentage increase of bacterial growth reduction via PAC with adsorbed SMZ compared to pure SMZ. The double arrow with y and y′ indicates the percentage increase of bacterial growth reduction via PAC with adsorbed G compared to pure G. The number next to the star indicates the contact time of PAC with antibacterial agent, expressed in minutes.

**Figure 6 pathogens-10-01066-f006:**
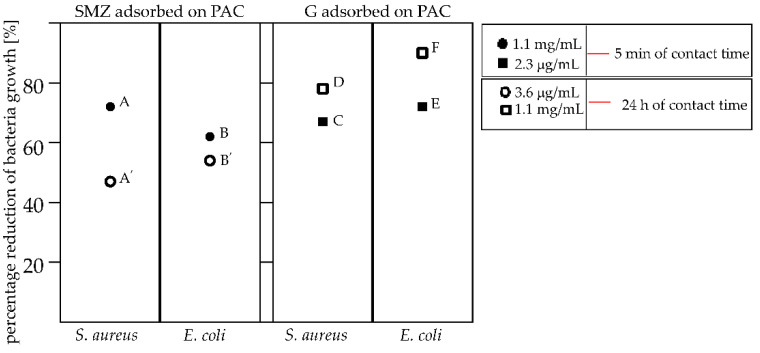
Percentage reduction of the bacteria, including the contact time of PAC with the adsorbed antibacterial agents. The application of PAC with the adsorbed SMZ: the 5 min of contact time, corresponding to 1.1 mg/mL of SMZ toward *S. aureus* (**A**) and *E. coli* (**B**) and the 24 h of contact time, corresponding to 1.1 mg/mL of SMZ toward *S. aureus* (**A′**) and *E. coli* (**B′**). The application of PAC with the adsorbed G: the 5 min of contact time, corresponding to 2.3 µg/mL of G toward *S. aureus* (**C**), the 24 h of contact time, corresponding to 3.6 µg/mL of G toward *S. aureus* (**D**), the 5 min of contact time, corresponding to 2.3 µg/mL of G toward *E. coli* (**E**) and the 24 h of contact time, corresponding to 3.6 µg/mL of G toward *E. coli* (**F**).

**Figure 7 pathogens-10-01066-f007:**
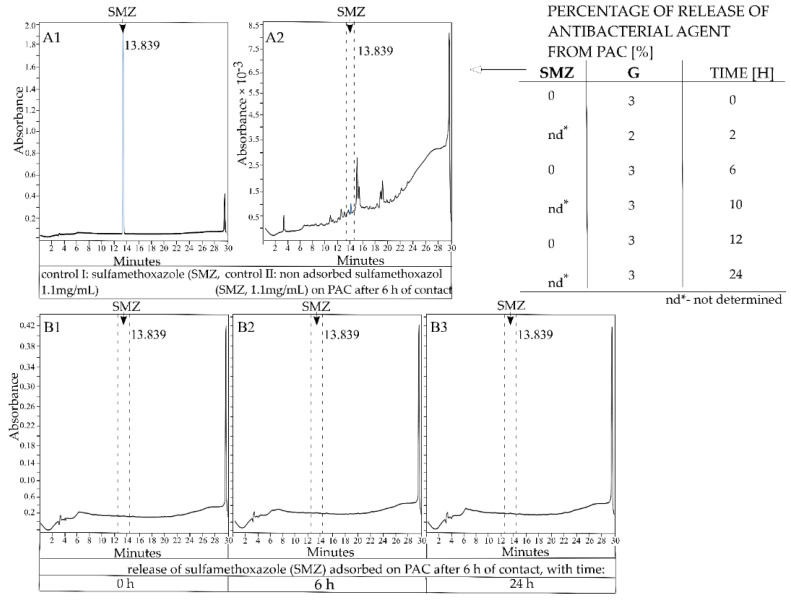
The release of gentamycin sulfate and sulfamethoxazole from PAC under aqueous conditions. Left: HPLC chromatogram images of (**A1**) sulfamethoxazole in a 1.1-mg/mL concentration and (**A2**) non-adsorbed sulfamethoxazole on PAC after 6 h of contact, and the release of sulfamethoxazole absorbed on PAC (6 h contact) over (**B1**) 0 h, (**B2**) 6 h, and (**B3**) 24 h. Right: The percentage release of gentamycin sulfate and sulfamethoxazole from PAC under aqueous conditions (pH 7.4).

**Table 1 pathogens-10-01066-t001:** Priority pathogen list adjusted to the WHO recommendations [6].

Pathogen List	Antibiotic- Resistant	Priority
*Acinetobacter baumannii*	carbapenem	CRITICAL
*Pseudomonas aeruginosa*	carbapenem	
*Enterobacteriaceae* *	carbapenem; 3rd-generation cephalosporin	
*Enterococcus faecium*	vancomycin	HIGH
*Staphylococcus aureus*	methicillin; vancomycin	
*Helicobacter pylori*	clarithromycin	
*Campylobacter*	fluoroquinolone	
*Salmonella* spp.	fluoroquinolone	
*Neisseria gonorrhoeae*	3rd-generation cephalosporin, fluoroquinolone	
*Streptococcus pneumoniae*	penicillin	MEDIUM
*Haemophilus influenzae*	ampicillin	
*Shigella* spp.	fluoroquinolone	

* *Enterobacteriaceae* include: *Klebsiella pneumonia* and *Escherichia coli*, *Enterobacter* spp., *Serratia* spp., *Proteus* spp., *Providencia* spp., and *Morganella* spp.

**Table 2 pathogens-10-01066-t002:** Values of the physical parameters of PAC.

Parameter	Analysis Technique	Value
Moisture	drying method	2.82 ± 0.22% analytical
Ash content	oven method	0.78 ± 0.03% analytical 0.80 ± 0.03% dry
Volatile matter	oven method	3.20 ± 0.03% analytical 3.32 ± 0.03% dry ash-free
pH_PZC_	Glass–electrode method	7.010
Carbon content	X-ray spectrophotometry	93.6% weight
Specific surface area (BET)	benzene adsorption	1267 m^2^/g
Micropore volume	thermogravimetric	0.499 cm^3^/g
Mesopore volume	thermogravimetric	0.205 cm^3^/g
Average width of mesopores	thermogravimetric	4.69 nm

## Data Availability

The data presented in this study are available on request from the corresponding author. The data are not publicly available due to institutional restriction and confidentiality.

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
