# Peer review of "Antibacterial Agents Adsorbed on Active Carbon: A New Approach for S. aureus and E. coli Pathogen Elimination"

_pathogens, 2021, doi:10.3390/pathogens10081066_

Round 1
Reviewer 1 Report
Dear authors, I read with interest your paper. The design and content did not convince me to recommend your paper for publication in this journal that has a high standard. Here are my reasons:
- the novelty of the study is limited
- additional experiments needed to prove the efficency of PAC against bacterial inhibition (such as SEM images, Live/Dead Bacterial viability )
- Figure 5 and Figure 6-difficult for readers to follow the informations, legends is missing
- a detailed description of the mechanism involved is missing
Reviewer 2 Report
The article by Burchacka et al. evaluates the antibacterial activity of pharmaceutical activated carbon (PAC) in association with adsorbed antibiotics gentamycin and sulfamethoxazole against S. aureus and E. coli. This formulation is tested in order to determine its potential as an antibacterial component of bandages in the future. Sulfamethoxazole (1.1 mg/mL) retained growth of S. aureus and E. coli at 20.5 and 26.5%, respectively, whereas sulfamethoxazole adsorbed on PAC increased the reduction of tested bacteria in the range of 47 to 72%. The use of PAC with adsorbed gentamycin (3.6 mg/mL) improved the effectiveness of E. coli growth reduction by 50% compared to the pure antibiotic. The research design and the methods were adequate and the results are of interest to the research field. The manuscript is recommended for publication but minor corrections should be performed in the text:
- Please review the text because there are some grammar and English typos.
- Please check the information about mesopores and micropores in Table 1, Figure 1, and text because the information described is in conflict.
- Please complete the references. Most of them are incomplete (without volume, issue, and pages).
Round 2
Reviewer 1 Report
Dear authors, the revised version of the manuscript has not been sufficiently improved to warrant publication in Pathogens.